# Data-dependent Gaussian Prior Objective for Language Generation

**Zuchao Li**[1,2,3], **Rui Wang**[4,*], **Kehai Chen**[4], **Masao Utiyama**[4], **Eiichiro Sumita**[4],
**Zhuosheng Zhang**[1,2,3] , **Hai Zhao**[1,2,3,*]
[1]Department of Computer Science and Engineering, Shanghai Jiao Tong University
[2]Key Laboratory of Shanghai Education Commission for Intelligent Interaction
and Cognitive Engineering, Shanghai Jiao Tong University, Shanghai, China
[3]MoE Key Lab of Artificial Intelligence, AI Institute, Shanghai Jiao Tong University
[4]National Institute of Information and Communications Technology (NICT), Kyoto, Japan
`charlee@sjtu.edu.cn, zhangzs@sjtu.edu.cn, zhaohai@cs.sjtu.edu.cn,`
`{wangrui, khchen, mutiyama, eiichiro.sumita}@nict.go.jp`

## Abstract

For typical sequence prediction problems such as language generation, maximum likelihood estimation (MLE) has commonly been adopted as it encourages the predicted sequence most consistent with the ground-truth sequence to have the highest probability of occurring. However, MLE focuses on once-to-all matching between the predicted sequence and gold-standard, consequently treating all incorrect predictions as being equally incorrect. We refer to this drawback as *negative diversity ignorance* in this paper. Treating all incorrect predictions as equal unfairly downplays the nuance of these sequences' detailed token-wise structure. To counteract this, we augment the MLE loss by introducing an extra Kullback–Leibler divergence term derived by comparing a data-dependent Gaussian prior and the detailed training prediction. The proposed data-dependent Gaussian prior objective (D2GPo) is defined over a prior topological order of tokens and is poles apart from the data-independent Gaussian prior (L2 regularization) commonly adopted in smoothing the training of MLE. Experimental results show that the proposed method makes effective use of a more detailed prior in the data and has improved performance in typical language generation tasks, including supervised and unsupervised machine translation, text summarization, storytelling, and image captioning.

## 1 Introduction

Language understanding is the crown jewel of artificial intelligence. As the well-known dictum by Richard Feynman states, "what I cannot create, I do not understand." Language generation therefore reflects the level of development of language understanding. Language generation models have seen remarkable advances in recent years, especially with the rapid development of deep neural networks (DNNs). There are several models typically used in language generation, namely sequence-to-sequence (seq2seq) models (Kalchbrenner & Blunsom, 2013; Sutskever et al., 2014; Bahdanau et al., 2015; Luong et al., 2015; Vaswani et al., 2017), generative adversarial networks (GANs) (Goodfellow et al., 2014), variational autoencoders (Kingma & Welling, 2013), and auto-regressive networks (Larochelle & Murray, 2011; Van Oord et al., 2016). Language generation is usually modeled as a sequence prediction task, which adopts maximum likelihood estimation (MLE) as the standard training criterion (i.e., objective). MLE has had much success owing to its intuitiveness and flexibility. However, sequence prediction has encountered the following series of problems due to MLE.

---

\* Corresponding author. Zuchao Li and Zhuosheng Zhang were internship research fellows at NICT when conducting this work. Hai Zhao was partially supported by Key Projects of National Natural Science Foundation of China (No. U1836222 and No. 61733011). Rui Wang was partially supported by JSPS grant-in-aid for early-career scientists (19K20354): "Unsupervised Neural Machine Translation in Universal Scenarios" and NICT tenure-track researcher startup fund "Toward Intelligent Machine Translation".

- Exposure bias: The model is not exposed to the full range of errors during training.
- Loss mismatch: During training, we maximize the log-likelihood, whereas, during inference, the model is evaluated by a different metric such as BLEU or ROUGE.
- Generation diversity: The generations are dull, generic (Sordoni et al., 2015; Serban et al., 2016; Li et al., 2016a), repetitive, and short-sighted (Li et al., 2016b).
- Negative diversity ignorance: MLE fails to assign proper scores to different incorrect model outputs, which means that all incorrect outputs are treated equally during training.

A variety of work has alleviated the above MLE training shortcomings apart from negative diversity ignorance. Negative diversity ignorance is a result of unfairly downplaying the nuance of sequences' detailed token-wise structure. When the MLE objective compares its predicted and ground-truth sequences, it takes a once-for-all matching strategy; the predicted sequence is given a binary label, either correct or incorrect. However, these incorrect training predictions may be quite diverse and letting the model be aware of which incorrect predictions are more incorrect or less incorrect than others may more effectively guide model training. For instance, an *armchair* might be mistaken with a *deckchair*, but it should usually not be mistaken for a *mushroom*.

To alleviate the issue of the negative diversity ignorance, we add an extra Gaussian prior objective to augment the current MLE training with an extra Kullback–Leibler divergence loss term. The extra loss is computed by comparing two probability distributions, the first of which is from the detailed model training prediction and the second of which is from a ground-truth token-wise distribution and is defined as a kind of data-dependent Gaussian prior distribution. The proposed data-dependent Gaussian prior objective (D2GPo) is then injected into the final loss through a KL divergence term. The D2GPo is poles apart from the commonly adopted data-independent Gaussian prior (L2 regularization) for the purpose of smoothing the training of MLE, which is also directly added into the MLE loss. Experimental results show that the proposed method makes effectively use of a more detailed prior in the data and improves the performance of typical language generation tasks, including supervised and unsupervised machine translation, text summarization, storytelling, and image captioning.

## 2 RELATED WORK

Natural language generation (NLG) has long been considered the most challenging natural language processing (NLP) task (Murty & Kabadi, 1987). NLG techniques have been widely adopted as a critical module in various tasks, including control-free sentence or poem generation (Zhang & Lapata, 2014) and input-conditioned language generation, such as machine translation, image captioning, text summarization, storytelling (Vaswani et al., 2017; Lample et al., 2018; Karpathy & Fei-Fei, 2015; Fan et al., 2018), and sentiment/tense-controlled sentence generation (Hu et al., 2017). In this work, we focus on input-conditioned language generation tasks, though our proposed method can also be applied to other language generation fields.

Input-conditioned language generation tasks are challenging because there is an information imbalance between the input and output in these tasks, especially for cases with non-text input (Shapiro, 1992). Reiter & Dale (2000) discussed different ways of building complicated knowledge-based systems for NLG. In recent years, neural networks (NNs), especially DNNs, have shown promising results in many NLP tasks. Bengio et al. (2003) first proposed the NN language model (NNLM) to exploit the advantages of NNs for language generation tasks. In an NNLM, the $n$-gram paradigm is extended by the generalization ability of NNs.

Given the ground truth sequence $s = \langle w_1, w_2, ..., w_{t-1} \rangle$, the NNLM adopts the equation

$$p_t \approx p(w_t|w_{t-n}, w_{t-n+1}, ..., w_{t-1}). \tag{1}$$

Mikolov et al. (2010) developed a more general implementation for a language model (called the recurrent NN language model (RNNLM) by integrating a Markov property using a recurrent NN (RNN) to address the NNLMs' theoretical inability to capture long-term dependencies:

$$p_t \approx p(w_t|\text{RNN}(w_1, w_2, ..., w_{t-1})). \tag{2}$$

The RNNLM is an effective solution because it is designed to capture long-term dependencies. Because of the vanishing gradient problem in RNNs, however, the long-term dependency processing

capability is limited. In contrast to an RNN, the Transformer (Vaswani et al., 2017) provides a new self-attention mechanism for handling long-term dependencies in text, resulting in robust performance across diverse tasks. Radford et al. (2018) proposed a Transformer language model called GPT, which uses a left-to-right architecture, where each token pays attention to previous tokens in the self-attention layers of the Transformer. Devlin et al. (2019) introduced a new pre-training objective: the masked language model (MLM), which enables the representation to fuse the left and right contexts and allows us to pre-train a deep bidirectional Transformer called BERT.

The generators of the most current language generation model use the RNNLM or Transformer language model structure. However, as pointed out by Bengio et al. (2015), fitting the distribution of observation data does not mean that satisfactory text will be generated, because the model is not exposed to the full range of errors during training. This is called the *exposure bias* problem. Reinforcement learning, GANs (Goodfellow et al., 2014; Yu et al., 2017), and end-to-end re-parameterization techniques (Kusner & Hernández-Lobato, 2016) have been proposed to solve this problem. The *exposure bias* is no longer an issue in reinforcement learning models because the training sequences are generated by the model itself.

Using MLE for the training objective leads to the problem of *loss mismatch*. Ranzato et al. (2015) incorporated the evaluation metric into the training of sequence-to-sequence (seq2seq) models and proposed the mixed incremental cross-entropy reinforce (MIXER) training strategy, which is similar to the idea of minimum risk training (Smith & Eisner, 2006; Li & Eisner, 2009; Ayana et al., 2016; Shen et al., 2016). MIXER uses decoder hidden states to predict the bias term and hence reduce the variance, while minimum risk training renormalizes the predicted probabilities. Zhang & Zhao (2018) introduced a new training criterion based on the Hellinger distance for the seq2seq model and empirically compared the models of two optimization categories: *minimum divergence* and *maximum margin*.

For the *generation diversity* problem, Serban et al. (2017) applied a latent variable hierarchical encoder–decoder dialog model to introduce utterance-level variations and facilitate longer responses. Zhao et al. (2017) presented a novel framework based on conditional variational autoencoders that improves generation diversity by sampling a latent variable $z$ and optionally adding linguistic features to constrain the style further.

There is an increasing interest in incorporating problem field knowledge in machine learning approaches (Taskar et al., 2004; Ganchev et al., 2010; Hu et al., 2016). One common way is to design specialized network architectures or features for specific knowledge (e.g., Liang et al. (2017; 2018)). In contrast, for structured probabilistic models, posterior regularization and related frameworks (Ganchev et al., 2010; Liang et al., 2009; Bellare et al., 2009) provide a general means to impose knowledge constraints during model estimation. Hu et al. (2018) established a mathematical correspondence between posterior regularization and reinforcement learning and, using this correspondence, expanded posterior regularization to learn knowledge constraints as the extrinsic reward in reinforcement learning. Our approach can be seen as incorporating a priori knowledge of the language field into language generation learning.

Additionally, Welleck et al. (2019) proposed a new objective, unlikelihood training, which forces unlikely generations to be assigned lower probability by the model. The difference is that (Welleck et al., 2019) focuses on low-frequency words while our model focuses on negative tokens. It is claimed that the likelihood objective itself is at fault, resulting in a model that assigns too much probability to sequences containing repetition and frequent words, unlike those from the human training distribution. From this point of view, there is a point in common with our motivation, which is to make the model prediction consistent with human training distribution to some extent.

## 3    BACKGROUND

Consider a conditional probability model for sequence predictions $\boldsymbol{y} \sim p_\theta(\boldsymbol{x})$ with parameters $\boldsymbol{\theta}$. The target sequence $\boldsymbol{y}$ can be conditioned on any type of source $\boldsymbol{x}$ (e.g., a phrase, sentence, or passage of human language or even an image), which is omitted for simplicity of notation. For the sequence $\boldsymbol{y} = \langle y_1, y_2, ..., y_l \rangle$, the probability $p_\theta(\boldsymbol{y}|\boldsymbol{x})$ is

$$p_\theta(\boldsymbol{y}|\boldsymbol{x}) = p_\theta(y_1|\boldsymbol{x})p_\theta(y_2|\boldsymbol{x}, y_1)...p_\theta(y_l|\boldsymbol{x}, y_{1:l-1}). \tag{3}$$

Commonly, sequence prediction models are trained using MLE (also known as teacher forcing) (Williams & Zipser, 1989). MLE minimizes the negative log-likelihood of $p_\theta(\boldsymbol{y}|\boldsymbol{x})$:

$$\mathcal{L}_{\textbf{MLE}}(\theta) = -\log p_\theta(\boldsymbol{y}|\boldsymbol{x}) = -\sum_{i=1}^{l} \log p_\theta(y_i|\boldsymbol{x}, \boldsymbol{y}_{<i}). \tag{4}$$

Optimizing the MLE objective $\mathcal{L}_{\textbf{MLE}}(\boldsymbol{\theta})$ is straightforward and meets the principle of empirical risk minimization while focusing on only minimizing losses of the correct target on the training data set.

However, there may be noise in the training data, and forcibly learning the distribution of a training set does not enable the obtained model to reach good generalization. Additionally, for sequence prediction, models trained subject to MLE cursorily evaluate all predictions as either correct or incorrect and ignore the similarity between the correct and "less incorrect" predictions. Incorrect predictions might range from nearly perfect (i.e., one token is mistaken with a synonym) to completely wrong, having nothing in common with the gold sequence. However, MLE training treats all incorrect training predictions equally, which implies that MLE fails to accurately assign scores to diverse (especially negative) model predictions.

## 4 D2GPO: DATA-DEPENDENT GAUSSIAN PRIOR OBJECTIVE

To capture the diversity of negative training predictions, we augment the MLE objective of the model with an additional objective $\mathcal{O}$ that more accurately models such a negative diversity. Without loss of generality, supposing $\tilde{\boldsymbol{y}}$ is the prediction candidate, we introduce a general evaluation function $f(\tilde{\boldsymbol{y}}, \boldsymbol{y}) \in \mathbb{R}$ independent of the model prediction, such that with a golden target token $y^*$, a higher $f(\tilde{y}, y^*)$ value indicates a better $p_\theta(\tilde{y}|\boldsymbol{x})$ for a target candidate $\tilde{y} \in V$ (where $V$ is the target candidate set). Note that $f(\tilde{\boldsymbol{y}}, \boldsymbol{y})$ can also involve other factors such as latent variables and extra supervisions.

There are two main methods of learning $f(\tilde{\boldsymbol{y}}, \boldsymbol{y})$ in the model. If $p_\theta$ is a GAN-like implicit generative model or an explicit distribution that can be efficiently reparametrized (e.g., Gaussian) (Kingma & Welling, 2013), then one effective method is maximizing $\mathbb{E}_{p_\theta}[f(\tilde{\boldsymbol{y}}, \boldsymbol{y})]$. The other method is computing the gradient $\nabla_\theta \mathbb{E}_{p_\theta}[f(\tilde{\boldsymbol{y}}, \boldsymbol{y})]$ using the *log-derivative* trick that can suffer from high variance but is often used for the large set of non-parameterizable explicit distributions.

Corresponding to the probability distribution of model predictions $p_\theta(\cdot)$, we define a prior distribution $q(\boldsymbol{y})$ (for each target $y_i$, it has its own unique distribution of $q_i = q(y_i)$) which is extracted and derived from the ground-truth data (e.g., language text in language generation tasks). To guide the probability distribution of model predictions $p_\theta(\cdot)$ to match the prior probability distributions $q(\cdot)$, we adopt Kullback–Leibler divergence. Considering also the learning of the evaluation function $f(\tilde{\boldsymbol{y}}, \boldsymbol{y})$, the loss for objective $\mathcal{O}$ is calculated as

$$\mathcal{L}_{\mathcal{O}}(\boldsymbol{\theta}, q) = KL(q(\boldsymbol{y})\|p_\theta(\boldsymbol{y}|\boldsymbol{x})) - \alpha\mathbb{E}_q[f(\tilde{\boldsymbol{y}}, \boldsymbol{y})], \tag{5}$$

where $\alpha$ is a weight for the evaluation function learning term. We derive the prior distribution $q(\boldsymbol{y})$ from the ground-truth data (which is independent of model parameters $\theta$), and therefore $\mathbb{E}_q[f(\tilde{\boldsymbol{y}}, \boldsymbol{y})]=0$. Hence, Eq. (5) becomes

$$\mathcal{L}_{\mathcal{O}}(\boldsymbol{\theta}, q) = KL(q(\boldsymbol{y})\|p_\theta(\boldsymbol{y}|\boldsymbol{x})), \tag{6}$$

in which KL divergence can be expanded as

$$KL(q\|p_\theta) = \mathbb{E}_p(\log(\frac{q}{p})) = \sum_i q_i * \log(q_i) - \sum_i q_i * \log(p_i). \tag{7}$$

The final objective for learning the model is written as

$$\min_\theta \mathcal{L}_{\textbf{MLE}}(\boldsymbol{\theta}) + \lambda\mathcal{L}_{\mathcal{O}}(\boldsymbol{\theta}, q), \tag{8}$$

where $\lambda$ is the balancing hyperparameter. Because optimizing the original model objective $\mathcal{L}_{\textbf{MLE}}(\boldsymbol{\theta})$ is straightforward, in the following, we omit the discussion of $\mathcal{L}_{\textbf{MLE}}(\boldsymbol{\theta})$ and focus on the proposed $\mathcal{L}_{\mathcal{O}}(\boldsymbol{\theta}, q)$.

The prior probability distribution $q(y^*)$ on $y^*$ can be obtained from the evaluation function $f(\cdot, \cdot)$ with a softmax operation. To expose the mass of the distribution over the classes, Hinton et al. (2015) introduced a softmax temperature mechanism; therefore, the relationship between $q$ and $f(\tilde{\boldsymbol{y}}, \boldsymbol{y})$ is

$$q(y^*) = \frac{exp(f(\tilde{\boldsymbol{y}}, y^*)/T)}{\sum_j exp(f(\tilde{y}_j, y^*)/T)}, \tag{9}$$

where $T$ is a temperature parameter. When $T \to 0$, the distribution becomes a Kronecker distribution (and is equivalent to a one-hot target vector); when $T \to +\infty$, the distribution becomes a uniform distribution. The softmax operation always turns an evaluation function $f(\cdot, \cdot)$ into a form of probability distribution no matter the form of the original $f(\cdot, \cdot)$; thus, we only focus on $f(\cdot, \cdot)$.

To find a good evaluation function, we have to mine token-wise diversity for all $y^*$. Considering all token types $\tilde{y}_j$ in a vocabulary, with respect to each $y^*$, there exists a prior topological order $ORDER(y^*)$ among all the known tokens, in which $y^*$ is always ranked top priority. $f(\tilde{y}_j, y^*)$ can then be defined as a monotonic function over the corresponding topological order so that it has a maximal value only when the input is $y^*$ itself. Note that defining $f(\cdot, \cdot)$ in this way leads to the resulting $q$ also being monotonic over the corresponding topological order. Considering that $q$ is *a priori*, it is fixed throughout the learning process.

The remaining questions are about how to find a meaningful evaluation function $f(\cdot, \cdot)$ for the distribution $q$. In language generation tasks, we may conveniently take word embedding as the token representation, and let the embedding distance determine such an order $ORDER(y^*)$ for each $y^*$. In this work, we adopt the cosine similarity of pre-trained embeddings to sort the token (word/subword) order.

**Discussion**   For the evaluation function $f(\cdot, \cdot)$ of $q$, we adopt the Gaussian probability density function, though later we also present experimental results for other types of functions in an ablation study. As the adopted Gaussian prior used in the training objective is derived from a data-dependent token-wise distribution, we call it the data-dependent Gaussian prior objective (D2GPo). This objective is a big departure from the Gaussian prior commonly adopted for smoothing in MLE training (which we the data-independent Gaussian prior). The following briefly explains why we chose the Gaussian probability density function and how our D2GPo mathematically differs from the data-independent Gaussian prior.

The central limit theorem indicates that suitably standardized sums of independent random variables have an approximately normal distribution. Thus, any random variable that arises as the sum of a sufficiently large number of small random components can be modeled accurately using a normal distribution. Embedding has a linear additive property (e.g., *king - man + woman $\approx$ queen*). The additive property of embedding can be explained by inspecting the training objective (Mikolov et al., 2013). Each dimension of an embedding represents a potential feature of the token. Considering each potential feature as an independent random variable, the sum follows a Gaussian distribution centered on the correct vocabulary unit $y^*$ according to the linear additive property. We can therefore use a Gaussian distribution for the embedding-distance-determined order to effectively model the distribution $q(y^*)$. An overview of the concepts underlying D2GPo is illustrated in Appendix A.1.

The D2GPo in this paper is different from the data-independent Gaussian prior in machine learning optimization theory. We hypothesize and experimentally verify that the embedding feature extracted from the data obeys the Gaussian distribution. The distribution from the prior knowledge of language data is used as a soft target to guide the model language generation process using knowledge distillation. The Gaussian prior in the machine learning optimization theory assumes that each component in the parameter $\theta$ is subject to a zero-mean Gaussian prior distribution, which is equivalent to L2 regularization. In general, our Gaussian prior objective is to act on the guiding target probability, while the Gaussian prior in machine learning is applied to the selection of model parameters.

## 5   EXPERIMENTS AND RESULTS

This section describes the experimental evaluation of D2GPo on a variety of typical language generation tasks: neural machine translation (NMT), text summarization, storytelling, and image captioning. The hyperparameters in D2GPo and effect analysis are given in Appendix A.7.

## 5.1 EMBEDDING PRE-TRAINING

Our proposed D2GPo approach for experimental tasks requires either word embeddings or byte-pair-encoding (BPE) (Sennrich et al., 2016b) subword embeddings. We generated the pretrained embeddings using fastText (Bojanowski et al., 2017) with an embedding dimension of 512, a context window of size 5, and 10 negative samples. For NMT, fastText was applied to the concatenation of source and target language monolingual corpora, resulting in cross-lingual BPE subword embedding. For text summarization, we generated BPE subword embedding only on the English monolingual corpora, while for the storytelling and image captioning, we obtained the word embedding also on the English monolingual corpora.

## 5.2 SUPERVISED NMT

We evaluated the model on several widely used translation tasks: WMT14 English-to-German (EN–DE), English-to-French (EN–FR), and WMT16 English-to-Romanian (EN–RO)[1] tasks, which all have standard large-scale corpora for NMT evaluation. Owing to space limits, the data details are provided in Appendix A.3. The sentences were encoded using sub-word types based on BPE, which has a shared vocabulary of 40,000 sub-word units for all three tasks. We chose the Transformer NMT (Vaswani et al., 2017) model as our baseline. For the hyperparameters of the Transformer (base/big) models, we followed the settings used by Vaswani et al. (2017). The BLEU (Papineni et al., 2002) score with multi-bleu.pl was calculated during the evaluation.

| System | EN–DE | EN–FR | EN–RO | EN–RO + STD |
|---|---|---|---|---|
| Vaswani et al. (2017) (base) | 27.30 | 38.10 | - | - |
| Vaswani et al. (2017) (big) | 28.40 | 41.00 | - | - |
| Transformer (base) | 27.35 | 38.44 | 33.22 | 36.68 |
| **+ D2GPo** | **27.93++** | **39.23++** | **34.00+** | **37.11+** |
| Transformer (big) | 28.51 | 41.05 | 33.45 | 37.55 |
| **+ D2GPo** | **29.10+** | **41.77++** | **34.13+** | **37.92+** |

Table 1: Comparison with baseline and existing systems on supervised translation tasks. Here, "++/+" after the BLEU score indicates that the proposed method was significantly better than the corresponding baseline Transformer (base or big) at significance levels $p < 0.01/0.05$. "STD" represents synthetic training data from (Sennrich et al., 2016b).

In Table 1, we report the performance of our full model, the baseline, and existing systems. Our baseline model obtains results similar to those of Vaswani et al. (2017), the existing strong model used for these tasks. The results indicate that our method performed better than the strong baselines for all language pairs. Our model is not only an improvement on the translation model of large-scale training sets but also performs better for small-scale training sets. Refer to Appendix A.8 and A.9 for analysis of the low-resource scenario and generation diversity.

## 5.3 UNSUPERVISED NMT

For unsupervised machine translation, we also used the three language pairs EN–DE, EN–FR, and EN–RO as our evaluation targets. Note that the evaluation performed on EN–DE uses *newstest2016* instead of *newstest2014* to ensure the results are comparable with the results of other works; this is unlike supervised machine translation. We used the masked sequence to sequence the pre-training (MASS) model (Song et al., 2019) as our baseline. Following the practice of Song et al. (2019), we pretrained our model with a masked sequence-to-sequence pre-training (MASS) objective (without D2GPo) on EN, FR, DE, and RO monolingual data samples from WMT 2007–2018 News Crawl datasets that respectively cover 190M, 60M, 270M, and 10M sentences. We then fine-tuned the models on the same monolingual data using the back-translation cross-entropy loss (Lample et al., 2018) and our D2GPo loss. For the training dataset, we filtered out sentences longer than 175 words in length and jointly learned 60K BPE sub-word units for each language pair.

---

[1]The results for EN–RO are evaluated on the dataset with diacritics removed in the reference text.

| Method | EN–FR | FR–EN | EN–DE | DE–EN | EN–RO | RO–EN |
|---|---|---|---|---|---|---|
| Artetxe et al. (2017) | 15.13 | 15.56 | 6.89 | 10.16 | - | - |
| Lample et al. (2017) | 15.05 | 14.31 | 9.75 | 13.33 | - | - |
| Yang et al. (2018) | 16.97 | 15.58 | 10.86 | 14.62 | - | - |
| Lample et al. (2018) | 25.14 | 24.18 | 17.16 | 21.00 | 21.18 | 19.44 |
| XLM (Lample & Conneau, 2019) | 33.40 | 33.30 | 27.00 | 34.30 | 33.30 | 31.80 |
| MASS (Song et al., 2019) | 37.50 | 34.90 | 28.30 | 35.20 | 35.20 | 33.10 |
| MASS + **D2GPo** | **37.92** | **34.94** | **28.42** | **35.62** | **36.31** | **33.41** |

Table 2: BLEU score comparisons between MASS and previous methods of unsupervised NMT.

As shown in Table 2, D2GPo consistently outperformed MASS (the state-of-the-art baseline) on all unsupervised translation pairs. Meanwhile, the MASS and XLMsystems leverage large-scale monolingual pre-training, and the decoder (generator, language model) can still be improved by our D2GPo loss in the fine-tuning phase. This demonstrates the efficiency of the proposed method.

## 5.4 TEXT SUMMARIZATION

Text summarization is a typical language generation task that creates a short and fluent summary of the given long-text document. Song et al. (2019) fine-tuned the MASS pretrained model on the text summarization task and achieved state-of-the-art results. We chose this model as our baseline, maintained consistent pre-training, and used D2GPo loss for enhancements in the fine-tuning phase. We used the Annotated Gigaword corpus as the benchmark, as detailed in Appendix A.4. In the evaluation, ROUGE-1, ROUGE-2, and ROUGE-L (Lin, 2004) are reported.

| | **Model** | **ROUGE-1** | **ROUGE-2** | **ROUGE-L** |
|---|---|---|---|---|
| Supervised | RNN-based seq2seq | 35.50 | 15.54 | 32.45 |
| | Nallapati et al. (2016) | 34.97 | 17.17 | 32.70 |
| Semi-supervised | MLM pre-training (Song et al., 2019) | 37.75 | 18.45 | 34.85 |
| | DAE pre-training (Song et al., 2019) | 35.97 | 17.17 | 33.14 |
| | MASS pre-training (Song et al., 2019) | 38.73 | 19.71 | 35.96 |
| | MASS + **D2GPo** | **39.23** | **20.11** | **36.48** |

Table 3: Performance on the text summarization task

Our results for text summarization are listed in Table 3. We compared our +D2GPo with our baseline MASS, which is the current state-of-the-art model; +D2GPo consistently outperformed the baseline on all evaluation metrics. The models with a semi-supervised setting yielded a large-margin improvement relative to the model without any pre-training, which demonstrates that the supervised pre-training is effective in the text summarization task.

## 5.5 STORYTELLING

Storytelling is at the frontier of current language generation technologies; i.e., stories must maintain a consistent theme throughout and require long-distance dependency modeling. Additionally, stories require creativity and a high-level plot with planning ahead rather than word-by-word generation (Wiseman et al., 2017).

We used the hierarchical story generation model (Fan et al., 2018) (which is introduced in Appendix A.5) as our baseline to test the improvements of D2GPo for the storytelling task. To guarantee the single-variable principle, we added only the D2GPo loss to the story generation model. The prompt generation model is consistent with Fan et al. (2018).

For automatic evaluation, we measured the model perplexity on validation and test sets. Table 4 shows results obtained using D2GPo. It is seen that with the addition of D2GPo, the Conv seq2seq + self-attention model substantially improved the likelihood of human-generated stories and even outperformed the ensemble or fusion models without increasing the number of parameters. Perplexity was further reduced with the addition of the fusion mechanism. These results suggest that

| Model | Params | Valid Perplexity | Test Perplexity |
|---|---|---|---|
| GCNN LM | 123.4 M | 54.50 | 54.79 |
| GCNN + self-attention LM | 126.4 M | 51.84 | 51.18 |
| LSTM seq2seq | 110.3 M | 46.83 | 46.79 |
| Conv seq2seq | 113.0 M | 45.27 | 45.54 |
| Conv seq2seq + self-attention | 134.7 M | 37.37 | 37.94 |
| Ensemble: Conv seq2seq + self-attention | 270.3 M | 36.63 | 36.93 |
| Fusion: Conv seq2seq + self-attention | 255.4 M | 36.08 | 36.56 |
| Conv seq2seq + self-attention + **D2GPo** | 134.7 M | **35.56** | **35.74** |
| Fusion: Conv seq2seq + self-attention + **D2GPo** | 255.4 M | **33.82** | **33.90** |

Table 4: Perplexity on WRITINGPROMPTS.

D2GPo improves the quality of language generation greatly, especially in settings where there are fewer restrictions on story generation tasks.

## 5.6 IMAGE CAPTIONING

Image captioning is a task that combines image understanding and language generation. It continues to inspire considerable research at the boundary of computer vision and natural language processing. We elected to experiment with image captioning to verify the performance of D2GPo on a language generation model having diverse types of input.

In our experiments, we evaluated our model on an ablated baseline (top-down, as detailed in Appendix A.6) (Anderson et al., 2018) against prior work on the MSCOCO 2014 caption dataset (Lin et al., 2014), which has became the standard benchmark for image captioning. For validation of model hyperparameters and offline testing, we used Karpathy splits (Karpathy & Fei-Fei, 2015), which have been used extensively in prior work. SPICE (Anderson et al., 2016), CIDEr (Vedantam et al., 2015), METEOR (Denkowski & Lavie, 2014), ROUGE-L, and BLEU were used to evaluate the caption quality.

| | BLEU-1 | BLEU-4 | METEOR | ROUGE-L | CIDEr | SPICE |
|---|---|---|---|---|---|---|
| Att2in (Rennie et al., 2017) | - | 31.3 | 26.0 | 54.3 | 101.3 | - |
| Att2all (Rennie et al., 2017) | - | 30.0 | 25.9 | 53.4 | 99.4 | - |
| Baseline: Top-down | 74.5 | 33.4 | 26.1 | 54.4 | 105.4 | 19.2 |
| Baseline + **D2GPo** | **75.2** | **33.6** | **26.3** | **55.1** | **106.6** | **19.7** |
| Baseline + SCST | 77.8 | 34.4 | 26.6 | 56.1 | 114.3 | 19.9 |
| Baseline + SCST + **D2GPo** | **78.0** | **34.7** | **26.8** | **56.3** | **116.8** | **20.2** |

Table 5: Image caption performance on the MSCOCO Karpathy test split.

Table 5 summarizes the performance of our full model and the ResNet Top-down baseline in comparison with the existing strong Self-critical Sequence Training (SCST) (Rennie et al., 2017) approach on the test portion of the Karpathy splits. To ensure a fair comparison, results are only reported for models trained with standard cross-entropy loss (i.e., MLE). All results are reported for a single model with no fine tuning of the input ResNet model. Our ResNet baseline performs slightly better than the SCST models. After incorporating our proposed D2GPo loss, our model improves further across all metrics.

## 6 EVALUATION FUNCTION

According to the analysis in Section 4, for the embedding, we used the Gaussian probability density function as our evaluation function $f(\cdot)$; however, to evaluate the effectiveness of different evaluation functions, we changed the function and tested the performance changes on the supervised NMT

EN-DE task. We used the same experiment settings as described in Section 5.2 and compared the BLEU score changes on the test set, as listed in Table 6.

| Evaluation Function | BLEU | △ |
|---|---|---|
| Baseline | 27.35 | |
| Gaussian | 27.93 | 0.58 ↑ |
| Random | 26.34 | 1.01 ↓ |
| Linear | 27.45 | 0.10 ↑ |
| Cosine | 27.62 | 0.27 ↑ |

Table 6: Ablation study on our proposed D2GPo with different evaluation functions on the supervised NMT WMT14 EN-DE task, with the Transformer-base model.

The table shows that the performance of Gaussian density, linear, and cosine functions increased while the performance of the random function decreased. This shows that the distance information obtained from embedding can effectively guide the generation process. Among these functions, the Gaussian density function had the greatest improvement, which agrees with our analysis of the embedding features obeying the Gaussian distribution. We postulate that because the linear and cosine functions are rough approximations of the Gaussian density function, they perform similarly to the Gaussian density function.

## 7 CONCLUSION

This work proposed a data-dependent Gaussian prior objective (D2GPo) for language generation tasks with the hope of alleviating the difficulty of *negative diversity ignorance*. D2GPo imposes the prior from (linguistic) data over the sequence prediction models. D2GPo outperformed strong baselines in experiments on classic language generation tasks (i.e., neural machine translation, text summarization, storytelling, and image captioning tasks).

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

# A  APPENDIX

## A.1  CONCEPTS UNDERLYING D2GPO

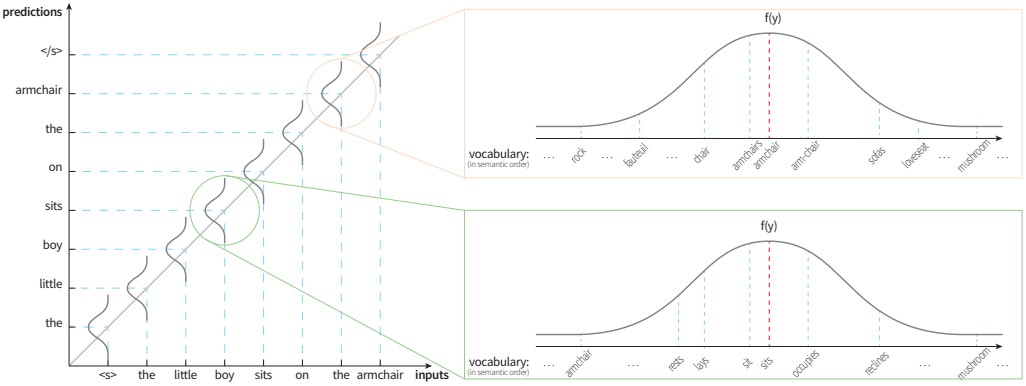

Figure 1: Overview of the concepts underlying D2GPo taking the example of the sentence *The little boy sits on the armchair*.

## A.2  TOPOLOGICAL ORDER

Specifically, for target $y^*$, we calculate the embedding cosine similarity as the distance $dist(\tilde{y}_j, y^*)$ of $y^*$ and all other token types in the vocabulary $\tilde{y}_j$, which are used to give the distance:

$$dist(\tilde{y}_j, y^*) = cosine\_similarity(emb(\tilde{y}_j), emb(y^*)). \tag{10}$$

Sorting by distance from small to large to obtain the topological order of token types yields

$$ORDER(y^*) = INDEX(sort([dist(\tilde{y}_1, y^*), dist(\tilde{y}_2, y^*), ..., dist(\tilde{y}_N, y^*)])). \tag{11}$$

where $N$ is the vocabulary size and $INDEX(\cdot)$ is used to obtain the new sequential index according to the distance sort. We define $ORDER(y^*)_j$ as the $j$-th value in $ORDER(y^*)$. Therefore, the evaluation function $f(\tilde{y}_j, y^*)$ is converted to a function defined on $ORDER(y^*)$:

$$f(\tilde{y}_j, y^*) = f(ORDER(y^*)_j). \tag{12}$$

For example, suppose the vocabulary has 5 tokens (i.e. N=5), for the golden target $y^*$ to be predicted, there is a relationship: $dist(\tilde{y}_2, y^*) < dist(\tilde{y}_3, y^*) < dist(\tilde{y}_1, y^*) < dist(\tilde{y}_5, y^*) < dist(\tilde{y}_4, y^*)$, the $ORDER(y^*)$ is $[3, 1, 2, 5, 4]$.

## A.3 SUPERVISED NMT DATA

For the EN–DE translation task, 4.43M bilingual sentence pairs from the WMT'14 dataset, which includes the Common Crawl, News Commentary, and Europarl v7 datasets, were used as training data. The *newstest2013* and *newstest2014* datasets were used as the dev set and test set, respectively.

For the EN–FR translation task, 36M bilingual sentence pairs from the WMT'14 dataset were used as training data. The *newstest2012* and *newstest2013* datasets were combined for validation and *newstest2014* was used as the test set, following the configuration of Gehring et al. (2017).

For the EN–RO task, we tested two settings; i.e., Europarl v7, which uses only the officially provided parallel corpus, and SETIMES2, which yields 600,000 sentence pairs for a low-resource supervised machine translation study. Alternatively, following the work of Sennrich et al. (2016a), we used synthetic training data (STD) of Sennrich et al. (2016a), which provides 2.8M sentence pairs for training. We used *newsdev2016* as the dev set and *newstest2016* as the test set. Our reported results on EN-RO are evaluated on a reference for which diacritics are removed from letters.

## A.4 TEXT SUMMARIZATION DATA

The Annotated Gigaword corpus (Napoles et al., 2012) was used as a benchmark (Rush et al., 2015). This data set is derived from news articles and comprises pairs of main sentences in the article (longer) and headline (shorter). The article and headline were respectively used as the source input sentence and reference. The data include approximately 3.8M training samples, 400,000 validation samples, and 2000 test samples.

## A.5 HIERARCHICAL STORY GENERATION MODEL

The hierarchical story generation model (Fan et al., 2018) was proposed for the situation in which a sentence called a *prompt* that describes the topic of the upcoming story generation is first generated, and then conditions on the *prompt* are applied when generating the story. Specifically, Fan et al. (2018) used a self-attention gated convolutional language model (GCNN) (Dauphin et al., 2017) as the sequence-to-sequence *prompt* generation model with top-$k$ random sampling. For *prompt*-to-story generation, they collected a dataset from Reddit's WRITINGPROMPTS forum in which each *prompt* has multiple story responses. With the dataset, they trained a story generation model that benefitted from a novel form of model fusion that improved the relevance of the story to the prompt and added a new gated multi-scale self-attention mechanism to model the long-range context.

## A.6 TOP-DOWN IMAGE CAPTION MODEL

The top-down image captioning model uses a ResNet (He et al., 2016) convolutional neural net pretrained on ImageNet (Deng et al., 2009) to encode each image. Similar to previous work (Rennie et al., 2017), the cited study encoded the full-sized input image with the final convolutional layer of Resnet-101 and used bilinear interpolation to resize the output to a fixed-size spatial representation of 10×10. This is equivalent to the maximum number of spatial regions used in our full model.

## A.7 Hyperparameters in D2GPo

During training with our D2GPo, the value of the standard deviation of the KL diversity item $\lambda$ was set to 0.1, and the softmax temperature was $T = 2.0$ in all experiments.

To study the effects of hyperparameters (i.e., the standard deviation $\lambda$ and softmax temperature $T$ in D2GPo) on the experimental results, we carried out experiments on WMT14 EN-DE with the Transformer-base model as the baseline [2] and set $\lambda$ as $[0, 0.1, 0.2, 0.5, 1.0]$, $T$ as $[1.0, 2.0, 5.0, 10.0]$.

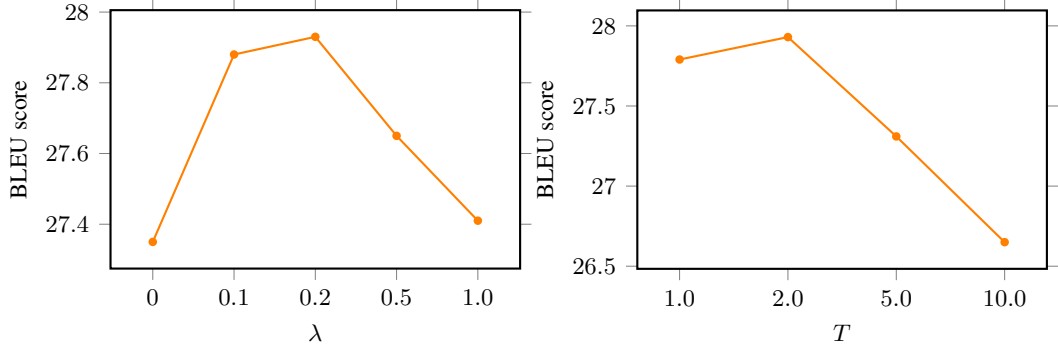

Figure 3: Performances on WMT14 EN-DE with different $\lambda$ values.
Figure 4: Performances on WMT14 EN-DE with different $T$ values.

The experimental results reveal that $\lambda$ affects the model training process. We believe that the reason is that a small value of $\lambda$ results in the model being unable to make full use of the prior knowledge (distribution), while a larger value of $\lambda$ will make the model more uncertain because of the higher probability of there being incorrect or even opposite words whose fastText embeddings are similar.

In addition, experimental results show that a small value of $T$ can improve the model to some extent, whereas a large value of $T$ will seriously decrease the performance of the model. Theoretically, when $T$ approaches infinity, the distribution $q$ becomes uniform, and there is no prior knowledge with which to guide the model. A loss penalty is applied to any model prediction, and an excessively high value of $T$ is thus harmful to training.

## A.8 D2GPo under a Low-resource Setting

Priors are generally more helpful in low-data regimes. We sampled 10,000, 100,000, and 600,000 paired sentences from the bilingual training data of WMT16 EN-RO to explore the performance of D2GPo in different low-resource scenarios. We used the same BPE code and learned fastText embeddings in all WMT16 EN-RO training data.

| Method | 10K | 100K | 600K |
|---|---|---|---|
| Baseline | 1.01 | 17.80 | 33.22 |
| **+ D2GPo** | 4.33 | 20.48 | 34.00 |

Table 7: Comparison of our baseline and our D2GPo method under different training data scales in terms of BLEU on the WMT16 EN-RO test set.

As shown in Table 7, D2GPo outperforms the baseline model, demonstrating the effectiveness of our method in low-resource scenarios. At the same time, the results show that the performance improvement provided by D2GPo increases with fewer training data. This shows that prior knowledge can substantially improve the performance of the model when training data are scarce. A possible reason is that the training data are insufficient to train a robust model. In this case, the injection of

---

[2]Owing to limited experimental resources and time, we only consider the situation that $\lambda$ and $T$ change separately; i.e., $T$ remains unchanged at 2.0 when $\lambda$ changes while $\lambda$ remains unchanged at 0.1 when $T$ changes.

prior knowledge can help train the parameters of the model and substantially improve the translation performance. However, with an increase in the number of training data, the model itself can be optimized well, and the improvement gained by introducing prior knowledge is not as substantial as before.

## A.9 EXAMPLES OF IMAGE CAPTIONING

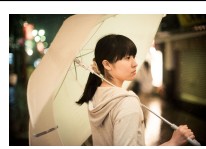
**Top-down:** a woman holding an umbrella in her hand
**+ D2GPo:** a woman is holding an umbrella
**+ SCST:** a woman holding an umbrella in a street
**+ SCST+ D2GPo:** a woman is holding an umbrella in the street

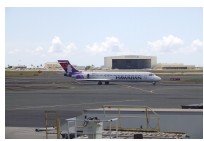
**Top-down:** a large airplane sitting on top of an airport runway
**+ D2GPo:** an airplane is sitting on top of an airport runway
**+ SCST:** a large jetliner sitting on top of an airport runway
**+ SCST+ D2GPo:** a large jetliner is sitting on top of an airport runway

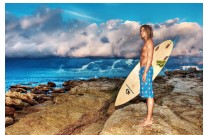
**Top-down:** a woman holding a surf board in the ocean
**+ D2GPo:** a woman is standing on the beach with a surfboard
**+ SCST:** a woman holding a surfboard on the beach
**+ SCST+ D2GPo:** a woman is standing on the beach with a surfboard

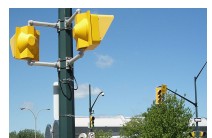
**Top-down:** a traffic light with a traffic light on it
**+ D2GPo:** a traffic light on the side of a traffic light
**+ SCST:** a yellow traffic light on the side of a street
**+ SCST+ D2GPo:** yellow traffic lights on the side of a street

Table 8: Captions generated for the left image by the various models described in the paper. The models trained with SCST return a more accurate and more detailed summary of the image. The models trained with D2GPo return a more grammatically complete sentence.

## A.10 ANALYSIS ON GENERATION DIVERSITY

Compared with traditional MLE training, D2GPo encourages negative diversity. To examine differences between D2GPo and MLE models, we counted high- and low-frequency words in the training set and compared the frequencies of low-frequency words predicted by the two models and the golden reference on the test set.

|  | #GOLD | Baseline | **+D2GPo** |
|---|---|---|---|
| #LF | 4915 | 3900 | 3998 |
| #SUM | 63086 | 55234 | 56129 |
| #RATIO | 7.79% | 7.06% | 7.12% |

Table 9: The statics of low frequency words in the reference and generations.

The experiment was carried out on WMT14 EN-DE, the baseline model was Transformer-base, and the statistics were calculated at the word level. We chose words with a frequency less than or equal to 100 in the training set as low-frequency words. We used the golden reference (#GOLD), baseline model prediction output, and +D2GPo model prediction output to count the total number of tokens (#SUM) and the number of low-frequency words (#LF). Results are given in Table 8.

The results show that compared with the baseline, the D2GPo optimized model generates more low-frequency words and has a higher ratio of low-frequency words. However, the number is still far

less than the golden reference . It is thus demonstrated that D2GPo increases the diversity of model output.

## A.11 EXAMPLES OF STORYTELLING

---

**Example Prompt:** The scientists have discovered something terrible .

---

*Generation with Conv seq2seq + Self-Attention* (PPL 37.37):

The discovery of the discovery was a long and boring process . It was a small , ⟨unk⟩ , and a simple and elegant solution . The scientists , scientists in charge , and the scientific community , was able to decipher the meaning of the universe .

The theory that the universe was not a very real one , was that the universe itself was not a planet , but a very , very real planet .

The discovery of a particular object in the universe that the universe was not . We found something that resembled something that could not be described , but something that could not be described .

We found a way to make the object a lot more specific . We found that it was a very , very large thing , a small , very small and very large .

It was the first thing we noticed that we found that it was a massive , large , enormous , enormous , giant , giant .

We found a way to make the whole thing.

We found that the entire

---

*Generation with Conv seq2seq + Self-Attention + D2GPo* (PPL 35.74):

" I don 't understand . Why did you tell us about it ? " The scientist 's assistant , Dr. harry , looked up from his tablet and back at the scientist .

" It 's ... you see. " He looked back up at Dr. harry . " What 's it mean ? "

" It means that the experiment is on the cusp of time . It 's not the same as it looks . There 's no explanation . It 's just that... "

" How is it possible that the experiment is going to fail ? " The scientist looked ut the window and then back at Dr. harry , who was still looking out the window , at the vast blackness of the room .

" You think they 'll do it ? "

" Well , no , we can 't be sure , Dr. harry . They 'll do it. "

Dr. harry turned to the scientist in front of him . " We can 't risk losing our best friend in the world

---

*Generation with Fusion of Conv seq2seq + Self-Attention* (PPL 36.08):

We thought that we should be able to see our entire species . The scientists were n't sure what we would find . But they had to be prepared for it . They could n't be ready .

The scientists had never seen anything like this , and yet they had n't seen anything like this ⟨unk⟩ in months . The scientists were just about to be in their own business , when they saw the results . They were n't ready for anything .

We thought that we should take a deep breath before we go . But it was n't .

There was n't any sign of any other intelligent life in the universe . It was nothing .

The scientists had no idea what was going on . They just watched , waiting for a signal .

They had no idea what it would mean .

But they could n't see us .

The scientists just knew . They were all afraid of what they would see . They could n't see the future .

They could n't see the future .

---

*Generation with Fusion of Conv seq2seq + Self-Attention + D2GPo* (PPL 33.82):

" I 'm afraid I wo n't be able to find out why my experiment is working . "

"Well , we 've been working on the project for about a month now . "

" It 's been a month and a half since I last saw it . "

" We 're all looking at the results . "

" You 've already been working on it for months now . You think we 've found that ? "

" I do n't know, but we do have a lot of research to do . "

" But it 's not like it was working , is it ? "

" We do n't know . We 're not looking for a breakthrough , it 's just an experiment . "

" It 's just an experiment ? People will die and the world may be destroyed . The disaster is about to happen, we have to act."

" What do you mean , it 'll not . It 's just an experiment . "

" No , no , no , it is something terrible we cannot ignore ."

---

Table 10: Example stories generated by the baselines and our full models.

