# OpenReview forum: "Data-dependent Gaussian Prior Objective for Language Generation"
_ICLR.cc/2020/Conference — Accept (Talk)_

### Official Review · AnonReviewer1 · 2019-10-22
**Official Blind Review #1**

**Rating:** 8

**Review:**

The paper introduces a new Gaussian prior objective, "D2GPo", that addresses the fact that in sequence generative models, all incorrect predictions are penalized equally by MLE, a phenomenon which the authors refer to as the negative diversity ignorance drawback. The proposed objective is simple to implement and can easily be added on top of a regular cross-entropy loss. The paper shows that the new objective shows consistent improvements across a wide range of tasks.

- Something that perturbed me a lot when reading Section 4, is that the function f should take as input 2 arguments, and not just 1. Typically, in Equation 7, from what I understand the numerator should be exp(f(y'_i, y_i)/T), and the denominator the sum over the exp(f(y'_j, y_i)/T). In other words, the value of f assigns a score to a word y'_j which depends on the target word y_i. This is also what suggests Figure 1: f depends on the target word. I think this should really be clarified in the paper, and f needs to be defined formally. In Figure 1, what is the exact value of f when the model outputs "armchair" / when the model outputs "armchairs". If I did not understand correctly, please point me out.

- The cross-entropy loss in a generative model is essentially the KL divergence loss KL(Q, P) where Q is the true distribution (i.e. a one-hot vector) and P is the model output. In Equation 7, when T -> 0, the distribution Q becomes this one-hot vector. In that case, from what I understand, the D2GPo objective is actually very similar to the initial MLE objective, except that you compute KL(P, Q) instead of KL(Q, P), and that Q is not exactly one-hot because you consider a T > 0. Can you confirm this? Also, any reason why you considered KL(P, Q) instead of KL(Q, P)?

- What value did you use for the temperature T? Did you study its impact?

- Same question for lambda in Equation 6, what is the value you considered, and did you study its impact?

- As mentioned in the introduction of the paper, one drawback with traditional neural generative, is the generation diversity, i.e. the fact that generations are generic, and an easy way to see this is to observe that generations are mostly composed of the most frequent words in the vocabulary. Did you evaluate whether a model trained with D2GPo has more diversity, and is more likely to generate rare words compared to a regular model trained with MLE only?

- Figure 1 in the appendix is helpful. It would be nice to move it to Section 4.

- In the related work, you write "the Transformer provides us with a more structured memory for handling long-term dependencies", which sounds a bit odd. There is no explicit memory component in the transformer, the ability to handle long-term dependencies comes instead from the self-attention mechanism.

Overall the paper tackles an interesting problem which I feel has received surprisingly little attention from the community. The paper is well written, and has a lot of experiments supporting the method. But I think Section 4 needs to be clarified, and some experimental details are missing. Also, more information about the impact of T and lambda (at least on one type of experiments) would be very useful to have.

**Experience Assessment:**

I have read many papers in this area.

**Review Assessment: Checking Correctness Of Derivations And Theory:**

I assessed the sensibility of the derivations and theory.

**Review Assessment: Checking Correctness Of Experiments:**

I assessed the sensibility of the experiments.

**Review Assessment: Thoroughness In Paper Reading:**

I read the paper at least twice and used my best judgement in assessing the paper.

---

> ### Author Response · Authors · 2019-11-13
> **Response to Reviewer #1**
>
> Thanks so much for your constructive reviews. Please see our response below.
>
> 1. About the incomplete definition problem.
>
> We are sorry for the reading discomfort caused by incomplete definition. Exactly, function f should take 2 arguments as input. When we wrote this expression, we focused on the situation given by y_i, so we made an incomplete expression. Thank you for your suggestion. We have modified this expression and formally define function f.
>
> 2. About the KL divergence item.
>
> This is a writing error, which we have fixed in this version. In machine learning, p is the distribution of real labels, and q is the distribution of predicted labels. While in this paper, p represents the predicted distribution of the model, q represents a real label distribution, so this error is caused. In the implementation, we use torch.nn.KLDivLoss()(input=p, target=q).
>
> 3. About the standard deviation (\lambda) and temperature T.
>
> In the experiments of this paper, we used the standard deviation of the KL diversity item  $\lambda$=0.1 and temperature  T=2.0. We have studied the influence of different values on the experimental performance on the supervised NMT EN-DE transformer-base model. The results are shown in Appendix A.7.
>
> 4. About generation diversity.
>
> We did a statistic on the generations and references (see Appendix A.9). According to the results, more low-frequency words were predicted. Therefore, we have a reason to believe that a model trained with D2GPo has more diversity, and is more likely to generate rare words compared to a regular model trained with MLE only.
>
> 5. About the position of Figure 1.
>
> Due to the limited space, we put Figure 1 in the appendix in the current version. We will try to adjust the expression of the paper and move Figure 1 to a more appropriate position in the final version. Thank you for your suggestion.
>
> 6. About the misleading words in related work.
>
> In fact, this sentence is relative to RNN. What the intention is to express is that Transformer provides a better memory representation than RNN (relying on self-attention to extract implicit memory instead of an explicit memory component). In our updated version, we have removed these misleading words.

---

> > ### Comment · AnonReviewer1 · 2019-11-13
> > **Thank you for the update**
> >
> > I thank the authors for the detailed answer. The questions I had were all well addressed in the response and in the revised version of the paper. The new sections in the appendix are also interesting. As a result, I decided to increase my score.

---

### Official Review · AnonReviewer3 · 2019-10-23
**Official Blind Review #3**

**Rating:** 8

**Review:**

This paper proposes to add a prior/objective to the standard MLE objective for training text generation models. The prior penalizes incorrect generations/predictions when they are close to the reference; thus, in contrast with standard MLE alone, the training objective does not equally penalize all incorrect predictions. For the experiments, the authors use cosine similarity between fastText embeddings to determine the similarity of a predicted word and the target word. The method is tested on a comprehensive set of text generation tasks: machine translation, unsupervised machine translation, summarization, storytelling, and image captioning. In all cases, simply adding the proposed prior improves over a state-of-the-art model. The results are remarkable, as the proposed prior is useful despite the variety of architectures, tasks (including multi-modal ones), and models with/without pre-training.

In general, it is promising to pursue work in altering the standard MLE objective; changes to learning objective seem orthogonal to the modeling gains made in many papers (as evidenced by the gains the authors show across diverse models). This paper opens up several new directions, i.e., how can we impose even more effective priors? The authors show that it's effective to use a relatively simple fastText-based prior, but it's possible to consider other priors based on large-scale pre-trained language models or learned models. In this vein, a concurrent paper "Neural Text Generation with Unlikelihood Training" has also shown it effective to alter the standard MLE objective. I think it would be nice to discuss this paper and related works. Overall, I think the approach is quite general and elegant.

My main criticism is that the writing was unfocused or unclear at times. The intro discusses a variety of problems in generation, before explaining that the authors only intend to tackle one ("negative diversity ignorance"). It would have been more helpful to read more text in the intro that motivated the problem of negative diversity ignorance and the proposed solution. The second paragraph in the Discussion in Section 4 is rather ambiguous and hand-wavy. It would be nice to see the authors' intuition described more rigorously (i.e., explicitly describing in math how the cosine similarity score is used in the Gaussian prior, or describing in math how the central limit theorem is used). Some of the existing mathematical explanation in section 4 could be made simpler or more clear (the description of f(y) seems to be a distraction since it doesn't end up in the final loss).

I would have also appreciated more analysis. After reading the paper, I have the following questions (which the authors may be able to address in the rebuttal):
* Do off-the-shelf fastText embeddings work well? How important is it to train fastText embeddings on the data itself? If off-the-shelf embeddings worked well, that could make the method easier to use for others in practice.
* How does the gain in performance with D2GPo vary based on the number of training examples? Priors are generally more helpful in low-data regimes. If that is the case here as well, you might get even more compelling results on low-data tasks (all tasks attempted here are somewhat large-scale, as I understand)
* Qualitatively, do you notice any difference in the generations? How does the model make mistakes (are these more "semantic" somehow, i.e. swapping a different synonym in). Perhaps the gaussian prior has some failure modes, i.e., where it increases the probability of very incorrect/opposite words because they have a similar fastText representation. These kinds of intuitions would be useful to know

I also have one technical question:
* When you compare against MASS (Song et al. 2019), do you use the same code and/or pre-trained weights from MASS, or do you pre-train from scratch using the procedure from MASS? (The wording in the text is somewhat ambiguous.) I'm just wondering how comparable the results are vs. MASS, or if it would be useful to know how your version of the pre-trained model does.


Despite my above questions/concerns, I think the proposed method or its predecessors could provide improvements across a variety of text generation tasks, so I overall highly recommend this paper for acceptance.


**Experience Assessment:**

I have published one or two papers in this area.

**Review Assessment: Checking Correctness Of Derivations And Theory:**

I assessed the sensibility of the derivations and theory.

**Review Assessment: Checking Correctness Of Experiments:**

I assessed the sensibility of the experiments.

**Review Assessment: Thoroughness In Paper Reading:**

I read the paper thoroughly.

---

> ### Author Response · Authors · 2019-11-13
> **Response to Reviewer #3**
>
> Thanks so much for your constructive reviews. Please see our response below.
>
> 1. About the reference to the related paper.
>
> The paper "Neural Text Generation with Unlikelihood Training" proposes a new objective, unlikelihood training, which forces unlikely generations to be assigned lower probability by the model. The paper claims that the likelihood objective itself is at fault, resulting in a model that assigns too much probability to sequences containing repeats and frequent words, unlike those from the human training distribution.  From this point of view, there are some similar points with our motivation, which are to some extent to make the model prediction consistent with human training distribution. The difference is that this paper focuses on low-frequency words, while our model focuses on negative tokens. Although from the final results, our model also reduces the prediction problem of low-frequency words. Therefore, these two works have some common ground, but there are also big differences. We have added a discussion to this paper in related work.
>
> 2. About the criticisms.
>
> Thank you for your valuable suggestions. We will revise the next version according to your suggestions due to the limited time in the rebuttal period.
>
> 3. About the question on off-the-shelf fastText embeddings.
>
> Yes, the off-the-shelf fastText embeddings work well for our proposed method. In our D2GPo for image captioning experiment, the off-the-shelf fastText embedding pretrained by Facebook \url{https://fasttext.cc/} is used due to the image captioning baseline system is performed on word-level. For other tasks, the reason why the off-the-shelf pretrained embedding is not used is that other tasks are based on the subword level, and the BPE codes for different experimental settings are different, resulting in the inconsistency of subword grain and vocabulary. Therefore, we pretrained the embedding on the training data (after BPE) to ensure the consistency of vocabulary (fastText embedding and model). However, we are also working to solve this problem. By modifying fasText's training objective, n-gram characters are not only used as feature input, but also as prediction objective. In this way, we hope to solve the problem of embedding input for different subword grain in the future.
>
> 4. About the question on the performance gain with D2GPo vary based on the number of training examples.
>
>
> Thank you for your reminding, we have added the experiment (see Appendix A.8) of the low resource setting of supervised NMT EN-RO during rebuttal. We sampled 10K, 100K, 600K paired sentences from the bilingual training data of WMT16 English-Romanian to explore the performance of our method in different low-resource scenarios. We used the same BPE codes learned in the pre-trained stage to tokenize the training sentence pairs and the same fastText subword embedding pretrained using all the 600K training data. The results show that too little training data is not enough to train a robust model. In this case, the injection of prior knowledge can help to train the parameters of the model and substantially improve the translation performance. However, with the increase of training data, the model itself can be optimized well, and the improvement by introducing prior knowledge is not as substantial as before.
>
> 5. About the question on generation quality and failure modes.
>
> We have made some observations on the generated text, but it's hard to say whether we can see the generation quality has changed directly, which is very subjective. As you said, our model will encourage negative diversity, so there must be some failure modes due to incorrect or even opposite tokens have similar fastText embedding. Therefore, we did a statistic on the generations and references (see Appendix A.9). In this statistic, we filtered low-frequency words in the training set and counted them in the generations and reference, and the results showed that more low-frequency words were predicted. Therefore, we have reason to believe that part of the improvements comes from the diversity brought by D2GPo.
>
> 6. About the technique question.
>
> We used the pretrained neural network weights provided by MASS and then used back translation to train the unsupervised NMT model. The training data and BPE code are consistent with MASS. At present, our D2GPo was only used in the back translation training phase. Because we don't have enough computation resources and time to replicate MASS's unsupervised NMT experiments, we reported his results directly. As for whether D2GPo is useful in the pretraining phase, we are trying to re-pretrain MASS. If there is an updated result, we will report it in the next version.

---

> > ### Comment · AnonReviewer3 · 2019-11-14
> > **Response to Rebuttal**
> >
> > These responses seem reasonable to me. A few further comments:
> >
> > 3. That makes sense to retrain the fastText embeddings because you're using subwords. Since fastText embeddings are character/n-gram based, can't you embed subwords directly though? (I'm not sure if it's common or effective to embed subwords directly with fastText though, just a thought.)
> >
> > 4. The low-resource experiment is very helpful - it does seem that the prior is more useful in low resource settings. This explains why the gains on e.g. machine translation are less than on other tasks.
> >
> > 5. The analysis on generation diversity is insightful. It's good to see that low frequency words are predicted more often - I think this experiment gives a better sense of where/when the method may be useful.
> >
> > 6. Great, yes it would be interesting to see if the objective is helpful during the pre-training phase.
> >
> > Overall, I am quite satisfied with the author rebuttal, so I will stand by my positive rating of the work.

---

> > > ### Author Response · Authors · 2019-11-15
> > > **Thank you very much for your constructive comments.**
> > >
> > > Thank you very much for your constructive comments.
> > >
> > > 3. As you mentioned, subword-level fastText embedding can be directly conducted in this work. Our future work is to propose a more general embedding method that can be applied to both subword-level and word-level.
> > >
> > > Thanks again.

---

### Official Review · AnonReviewer2 · 2019-10-24
**Official Blind Review #2**

**Rating:** 8

**Review:**

This paper introduces the use of data-dependent Gaussian prior, to overcome negative diversity ignorance problem that includes the exposure bias problem for sequence generation models. In addition to the usual MLE (teacher forcing) criteria, the authors add the KL divergence between the prediction and the Gaussian PDF on the word embedding space. Experimental results show that the proposed method consistently improves the performance of the state-of-the-art methods for neural machine translation, text summarization, storytelling, and image captioning.

I lean to accept this paper. The proposed method is well motivated and shown to be effective in several tasks for language generation.

I have some major comments about the evaluation function $f(\cdot)$. The authors propose to define it as a Gaussian distribution.
- While this choice seems to be reasonable, I would like to know how its standard deviation can be defined. If it is a hyperparameter, the sensitivity of different deviations for the performance should be experimentally reported. A small valued deviation would make the KL divergence close to zero, while a large one makes its convergence slow.
- Another way to remedy the problem of KL divergence above is applying Wasserstein distance instead of KL divergence. I would like to know if the authors have investigated the use.

Minor comments:
- White space should be inserted between "sequence-to-sequence" and "(seq2seq)" on the third page.
- If the authors define a sequence using a bold and italic font as $\boldsymbol{y}$, each token can be represented using an italic font to distinguish each token and the entire sequence: $\boldsymbol{y}=<y_1,...,y_l>$. Otherwise, the sequence can be defined as $\mathcal{Y}$ if the authors like to represent each token as a vector.
- There is a typo on the fifth page. The word "dada-independent" should be "data-independent."

**Experience Assessment:**

I have published in this field for several years.

**Review Assessment: Checking Correctness Of Derivations And Theory:**

I carefully checked the derivations and theory.

**Review Assessment: Checking Correctness Of Experiments:**

I carefully checked the experiments.

**Review Assessment: Thoroughness In Paper Reading:**

I read the paper thoroughly.

---

> ### Author Response · Authors · 2019-11-13
> **Response to Reviewer #2**
>
> Thanks so much for your constructive reviews. Please see our response below.
>
> 1. About the standard deviation (\lambda).
>
> We define the standard deviation of the KL diversity item as a hyperparameter, which is set to 0.1 in all experiments in this paper. We have studied the influence of different values on the experimental performance on the supervised NMT EN-DE transformer-base model. The results are shown in Appendix A.7. The experimental results show that the setting of  $\lamda$ will affect the training process of the model. If it is too large, the model will have more uncertainty and reduce the performance of the model. If it is too small, the enhancement effect is not obvious.
>
> 2. About the "distance" (difference) measure of distributions: f-divergence vs. Optimal Transport.
>
> There are two common ways to measure the distance between two distributions: f-divergence and Optimal Transport (OT). KL diversity is one of the f-divergence, and OT distance is also known as Wasserstein distance.
>
> OT is weaker than f-divergence in topology, which is very important in the generation model. Because the support set of data is often a low-dimensional manifold in the input space, there is probably no overlap between the real distribution and the generated distribution, resulting in the failure of the f-divergence due to it is a distance measure to capture the probability density ratio of distributions.
>
> Back to our work, the generation distribution of the model and the prior distribution of the data are defined on the same support set (i.e., vocabulary), so KL divergence is applicable.
>
> In our work, we fix the shape of the prior distribution, so we can use a more simple and intuitive distance measurement. In spite of this, we also try to implement the Wasserstein distance on our proposed D2GPo and used Sinkhorn Iteration to solve the problem. However, it is found that it is not suitable for language generation tasks due to the need to calculate the cost matrix. Suppose that for a batch, the size of the batch is B, the sequence length is S, and the vocabulary size is L, then for a batch, the target probability distribution size is (B x S) x L, and the cost matrix is (B x S) x L x L. For the language generation tasks, the target vocabulary is generally large (in supervised NMT EN-DE, the vocabulary size is 43640). In my preliminary experiment, the model tried to apply for 1783.30 GiB memory in the cost matrix calculation operation, so it is not feasible.
>
> 3. About the comments.
>
> For the space format and typo problems, we have fixed them in the updated version. For the expression of the formula, we will fix it according to your suggestion.

---

### Author Response · Authors · 2019-11-13
**Submission Update**

We thank all reviewers so much for the valuable comments on improving the quality of this work. We have updated the paper according to the feedback and our latest evaluations.

The revision primarily includes:
1. We fixed some typo and writing problems.
2. We added an introduction of D2GPo hyper-parameter and the analysis of its effect on translation performance (see Appendix A.7).
3. We explored the performance of D2GPo in low-resource scenarios and analyzed the possible reasons (see Appendix A.8).
4. We compared the model generation of D2GPo optimization and MLE optimization, and evaluated the diversity problem (see Appendix A.9).
5. We added related references and compared our work to relevant concurrent work.

---

### Comment · Area_Chair1 · 2019-11-14
**Reviewers 2 and 3, any further feedback?**

Dear Reviewers 2 and 3, thanks for your thoughtful input on this submission!  The authors have now responded to your comments.  Please be sure to go through their replies and revisions.  If you have additional feedback or questions, it would be great to get them this week while the authors still have the opportunity to respond/revise further.  Thanks!  And thanks to Reviewer 1 for doing this already!

---

### Decision · Program_Chairs · 2019-12-19

**Decision:**

Accept (Talk)

**Comment:**

This paper addresses the problem of poor generation quality in models for text generation that results from the use of the maximum likelihood (ML) loss, in particular the fact that the ML loss does not differentiate between different "incorrect" generated outputs (ones that do not match the corresponding training sequence).  The authors propose to train text generation models with an additional loss term that measures the distance from the ground truth via a Gaussian distribution based on embeddings of the ground-truth tokens.  This is not the first attempt to address drawbacks of ML training for text generation, but it is simple and intuitive, and produces improvements over the state of the art on a range of tasks.  The reviewers are all quite positive, and are in agreement that the author responses and revisions have improved the paper quality and addressed initial concerns.  I think this work will be broadly appreciated by the ICLR audience.  One negative point is that the writing quality still needs improvement.